# Assessing spatiotemporal variability in SARS-CoV-2 infection risk for hospital workers using routinely-collected data

**Jared K. Wilson-Aggarwal**[1], **Nick Gotts**[1], **Kellyn Arnold**[1], **Moira J. Spyer**[2,3], **Catherine F. Houlihan**[2,4], **Eleni Nastouli**[2,3¶], **Ed Manley**[1]*

**1** School of Geography, University of Leeds, Woodhouse, Leeds, United Kingdom, **2** Department of Clinical Virology, University College London Hospitals NHS Foundation Trust, London, United Kingdom, **3** Department of Infection, Immunity and Inflammation, UCL GOS Institute of Child Health University College London, London, United Kingdom, **4** Department of Infection and Immunity, University College London, London, United Kingdom

¶ Eleni Nastouli on behalf of the SAFER investigators (listed in S2 Appendix)
* E.J.Manley@leeds.ac.uk

**Data Availability Statement:** The data are available from UCLH but are not publicly available. Legal restrictions apply due to the potentially sensitive

## Abstract

The COVID-19 pandemic has emphasised the need to rapidly assess infection risks for healthcare workers within the hospital environment. Using data from the first year of the pandemic, we investigated whether an individual's COVID-19 test result was associated with behavioural markers derived from routinely collected hospital data two weeks prior to a test. The temporal and spatial context of behaviours were important, with the highest risks of infection during the first wave, for staff in contact with a greater number of patients and those with greater levels of activity on floors handling the majority of COVID-19 patients. Infection risks were higher for BAME staff and individuals working more shifts. Night shifts presented higher risks of infection between waves of COVID-19 patients. Our results demonstrate the epidemiological relevance of deriving markers of staff behaviour from electronic records, which extend beyond COVID-19 with applications for other communicable diseases and in supporting pandemic preparedness.

## Introduction

The rapid spread of the severe acute respiratory syndrome coronavirus 2 (SARS-CoV-2), that caused the COVID-19 pandemic, has challenged the resilience of healthcare systems. The need to protect front-line medical staff was quickly acknowledged, whereby healthcare workers (HCWs) were identified as three times more likely to test positive for COVID-19 than the general public [1]. While the global prevalence of infection in HCWs has been estimated at 11% [2], there was considerable variation in the early stages of the pandemic with one London hospital reporting infection in 44% of staff [3]. SARS-CoV-2 infection can be acquired by HCWs from their family and from the community, but they are also at risk of infection within the healthcare environment, where the modes of transmission are no different; aerosols, droplets and direct contact [4]. Protecting our front-line HCWs and patients by preventing

nature of the data and to protect the privacy of individuals. Data are however available from UCLH upon reasonable request and providing data security prerequisites are met. Data requests should be directed to the joint research office: uclh.randd@nhs.net.

**Funding:** EN, CH and EM were awarded funding from the UCLH/UCL NIHR Biomedical Research Centre and the UKRI MRC (grant ref: MC_PC_19082). Additional funds were awarded to EN from the UCLH Charity, and to EM from the Alan Turing Institute.

**Competing interests:** The authors have declared that no competing interests exist.

SARS-CoV-2 infection is a priority for hospitals, and requires an understanding of the risks associated with transmission.

In the community, the transmission dynamics of SARS-CoV-2 depend on the frequency and duration of contacts between infectious and susceptible individuals, which are somewhat determined by their mobility [5]. Community level interventions that focus on reducing the movements and contact rates of individuals have been successful in reducing transmission [6], as these social forces underpin transmission dynamics [7]. However, in the healthcare environment, similar interventions are less appropriate or practical as HCWs are required to have contact with patients. Nosocomial transmission of communicable diseases, such as SARS-CoV-2, is prevented through infection prevention and control (IPC) measures; that allow HCWs to safely conduct their work without the need to significantly reduce their within hospital mobility or patient contacts [8]. Examples of IPC measures in hospitals include the use of personal protective equipment (PPE), administrative controls (e.g. staff cohorting) and environmental controls (e.g. controlling air flow).

Surges in hospital admissions of COVID-19 patients resulted in stretched resources [9–11] made worse by staff shortages [12,13], both of which can compromise IPC activities. In these circumstances, the risk of infection for HCWs will depend not only on variations in the capacity to adhere to IPC policies, but also on the contact rates and mobility of individuals [7]. When events such as outbreaks and pandemics perturb the healthcare system in a way that negates IPC, there is a need to rapidly assess and monitor the risk of infection for staff.

During the early stages of the pandemic, risk factors for HCWs testing positive for COVID-19 included the lack of appropriate PPE [14,15], being of Black, Asian or minority ethnicity (BAME; [16]), working in doctor, nursing or healthcare assistant roles [2,14,17–22] and working night shifts [23]. There is also evidence for spatial variation in the risk of infection, whereby staff working on COVID-19 wards were more likely to test positive [2,17,19,20,22], but there have been relativley few investigations into how HCW mobility and patient interaction within the healthcare envirnonement influences the risk of infection; likely owing to the scarcity of data on HCW behaviour. To our knowledge, no studies have been conducted on HCW mobility, while studies on variaitons in patient engagement have contrasting results, with some finding higher risks for HCWs with more frequent contact with COVID-19 patients [14,17], and others finding either no difference or lower risks for those interacting with COVID-19 patients [15,24]. Intuitively, the current evidence suggests HCW behaviour in the workplace can determine their risk of infection, however, routinely collected data sources of HCW activity are underutilised, yet their inclusion in risk models could facilitate rapid risk assessments during disease outbreaks.

We have previously outlined how routinely collected hospital data, in the form of security door logs and electronic medical records, readily provide indicators for HCW behaviour within the hospital [25]. In this paper we investigate whether behavioural markers for HCW mobility and patient contacts are associated with the risk of individuals testing positive for COVID-19 in a London hospital during the first year of the pandemic. Distinct from previous studies, this investigation demonstrates a means to rapidly assess and monitor the risk of infection for all staff with evidence of activity in the hospital, while also providing insights into how risk varies between discrete spatial areas and in time.

## Methodology

### Study site and context

University College London Hospital (UCLH) is a tertiary teaching hospital located in central London. The main building is a 16 storey structure known as the Tower, which is linked to two other buildings; the Podium and the Elizabeth Garett Anderson Wing.

During the COVID-19 pandemic the UCLH Tower became a key COVID-19 hospital in London. We identified three stages during the first year of the pandemic between March 2020 and March 2021, using the daily number of COVID-19 patients in the hospital: (1) March 1st–June 30th 2020 (i.e. the 'first wave') when the first peak in COVID-19 admissions at the hospital was experienced and during which the WHO declared a pandemic (March 11th 2020); (2) July 1st–September 30th 2020 (i.e. the 'summer lull') when the number of COVID-19 patients in the hospital remained at a low level; and (3) November 1st 2020—March 31st 2021 (i.e. the 'second wave') when a subsequent peak of COVID-19 hospital admissions occurred and the mass-vaccination programme began (December 8th 2020). Data for the month of October 2020 were discarded, as records either could not be extracted or had an unusually low number of events (indicating an issue with extraction).

### Causal inference

Using observational data to infer causal relationships is notoriously challenging, and requires researchers to be explicit in their assumptions when conducting analyses [26–28]. To estimate the causal effect of a particular 'exposure' variable on an outcome of interest, it is necessary to remove (or adjust for) all hypothesised associations that confound the causal relationship. Directed acyclic graphs (DAGs) provide a formalised and rigorous framework for estimating causal effects, since they help to identify the covariates that must be adjusted for in statistical analyses and provide a transparent means for conveying a researcher's assumptions about the underlying data-generating process.

In this study, we aim to estimate the degree to which different factors affect the probability (likelihood) of HCWs testing positive for COVID-19 during each of the three identified stages of the pandemic. We adopt the formal framework provided by DAGs in order to estimate the total causal effect on the outcome probability for the following observed covariates (i.e. exposures) that were identified from the literature as influencing the risk of COVID-19 infection (Fig 1): age [29–31], ethnicity [16], job role [2,14,17–22,32], shift patterns [23,31,33], mobility and space use in the hospital [2,17,19,20,22], and patient contacts [14,17]. Crucially, two assumptions are made: (a) that a HCWs level of patient engagement is determined by their role and shifts, and (b) that their mobility and space use is a product of the patients they are required to see. In the supporting information we provide the adopted DAG with notations (S1 Appendix), detailing assumptions and justifications for hypothesised relationships between variables, and a note on potential unobserved confounders.

### Data sources & processing

For the duration of the study UCLH staff had access to a staff testing programme that included testing of combined nose/throat swabs for SARS-CoV-2 RNA by rtPCR: in the first wave this was for symptomatic staff and after May 2020 it included weekly testing of asymptomatic staff. Positive and negative SARS-CoV-2 PCR test results were extracted from the hospital's electronic health record system. Data fields included the test result, a pseudonymous identifier for the individual and the datetime for the test. The age, ethnicity and role of staff were extracted from electronic staff records. Ethnicities were categorised into either BAME or white. Staff

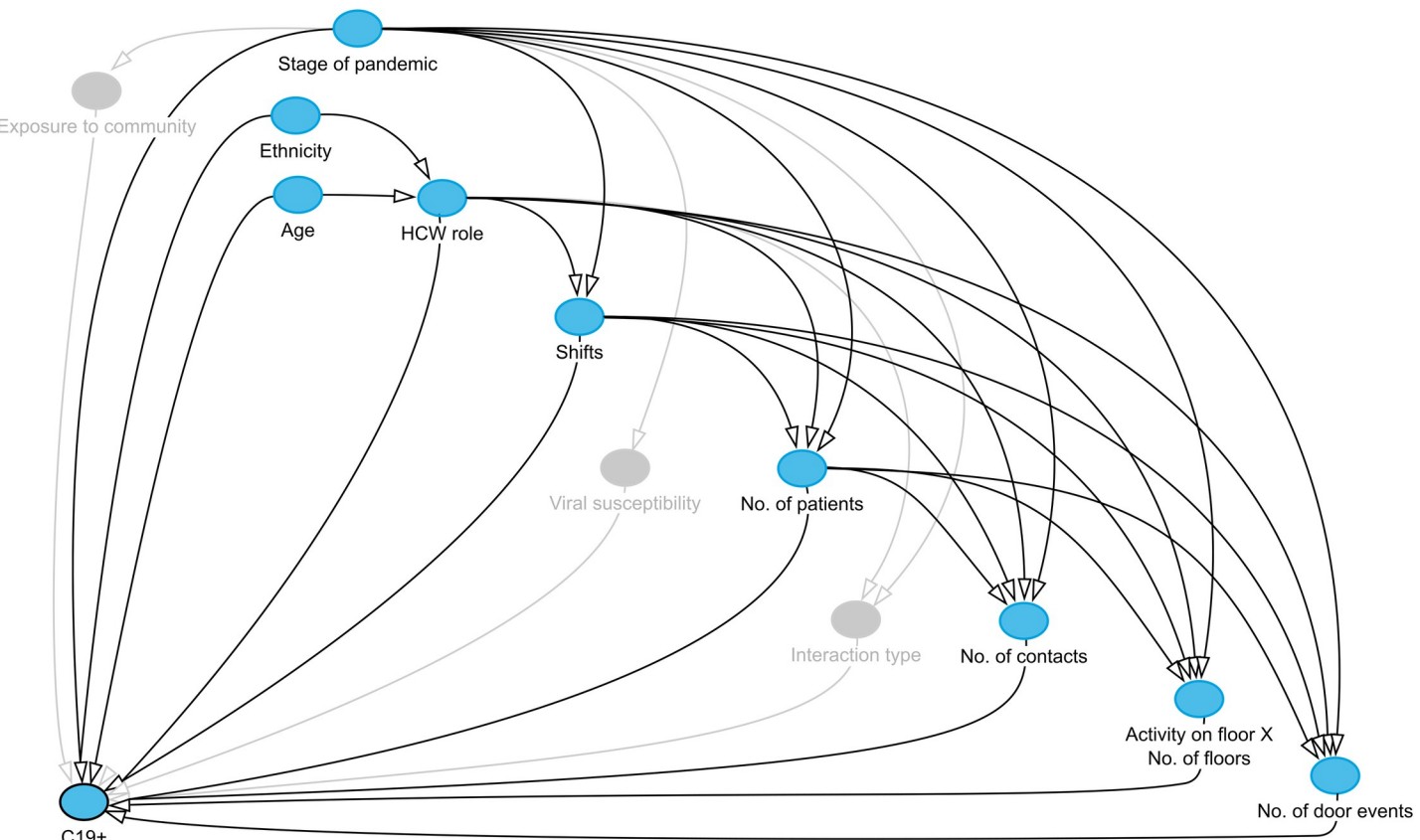

**Fig 1. Directed acyclic graph (DAG).** The DAG depicts the hypothesised relationships between variables considered to influence the probability of healthcare workers testing positive for COVID-19 (C19+). The blue nodes represent the observed variables and the grey nodes represent unobserved variables. Arrows connecting nodes show the (directional) relationship between variables.

roles were categorised into admin, allied health professionals, 'doctor: consultants', 'doctor: trainee', 'doctor: other' (e.g. general medical practitioner), porters, cleaners, healthcare assistants, nurses, physiotherapists and other clinical (e.g. pharmacist, phlebotomist and ambulance care assistant).

The date of each COVID-19 test was used to extract the individual's security door logs and patient contacts two weeks prior to the test. The two-week time period was determined based on the incubation of COVID-19, which can be up to 14 days [34], and with the assumption that an individual's behaviour (within hospital mobility and patient contacts) during this time period will best reflect their risk of testing positive. While we acknowledge that the actual exposure of HCWs to the virus is not known (e.g. due to contact with the virus and infectious individuals in the community/household, and other unrecorded contacts within the hospital), these metrics can provide a proxy for within-hospital exposure. Models using metrics derived from data 7 days and 2 days prior to a test were also performed to test the robustness of the model we adopted; the results of which are provided in the supplementary results (S1 Appendix). While the results are similar between models, we argue the model using 14 days of data is more appropriate due to the variation in incubation times of the virus, and the ability to include data on individuals with less frequent shifts.

Data sources and data processing for metrics of HCW behaviour are described elsewhere [25]. Briefly, using the hospital's electronic medical record system, we extracted the face to face

patient contact events for all HCWs and calculated the total number of patients and the total number of patient contact events. Door events were taken from the security door access logs and used to calculate an individual's mobility as inferred from the total number of door events. Both the door access logs and patient contacts were used to determine the total number of floors HCWs were active on. The aforementioned metrics were also calculated separately for a subset of the data involving only events relating to COVID-19 i.e. contacts with COVID-19 positive patients or activity on COVID-19 floors. COVID-19 floors were identified as those that handled a large share (>15%) of COVID-19 patients during the entire observation period; floors 1, 3, 7, 8, 9 and 10. Lastly, we determined whether HCWs had evidence of activity on specific floors, treating each floor as a binomial variable (1/0).

Staff rostering was not available electronically or on a centralised system for all HCW roles. Therefore, to prevent the exclusion of key staff groups and to allow the inclusion of data on shift patterns, we inferred the number of shifts worked in the Tower building from both logged door events and patient contacts. A new shift was identified by a temporal gap between events that was at least seven and a half hours. A night shift was determined by the time of the first event, whereby an event between 5pm and 5am identified a night shift. We use the available roster data to conduct an analysis to validate these methods and present the results in the supplementary methods (S1 Appendix). Specifically, we investigate the accuracy of using either a 4 hr, 7.5 hr or 11 hr temporal gap to identify distinct working rostered shifts. We found that using a 7.5 hr gap was the most effective method of correctly identifying shifts with the fewest errors.

We focused the analysis on HCWs who had activity in the Tower building as this was where the majority of COVID-19 patients were handled. Therefore test results for individuals with no evidence of activity (patient contacts and door events) in the Tower building were excluded from the analysis. Test results for individuals with erroneous shift metrics were also excluded; day shifts with more than one date associated with them or night shifts with more than two dates, and when the total number of shifts was greater than 14. We also excluded all test results for individuals after they had had a positive test.

## Statistical analysis

Mixed-effect logistic regression models were used to investigate the probability of HCWs testing positive for COVID-19. Statistical analyses were conducted in R [35] and the models were built using the 'lme4' package (v1.1–18.1). All models included individual ID as a random effect; the covariates included in the model as fixed effects were determined for each exposure individually according to the hypothesised DAG (Fig 1 and S2 Table in S1 Appendix). When included in models, the number of patients, number of patient contacts and the number of door events were all log transformed (base 2). We allowed for the effects to vary across time by including an interaction term with stage of the pandemic. For post hoc comparisons the package 'emmeans' (v1.3.3) was used to estimate p-values, adjusted odds ratios (OR) and 95% confidence intervals (CI). A Bonferroni correction was applied for comparisons among and between groups.

## Ethics statement

In this retrospective study no interventions were conducted. No consent was acquired and data were de-identified prior to analysis. The study protocol was approved by the NHS Health Research Authority (South Central–Berkshire REC ref 20/SC/0147, protocol number 130861) and ethical oversight was provided by the UCLH research ethics committee (IRAS project ID: ref. 281836). UCL GOS ICH R&D approval number 20PL06.

## Results

We analysed data for HCWs that had submitted the result of a COVID-19 test to the hospital testing programme between March 2020 and March 2021, and that had logged door events and/or patient contacts in the Tower building at UCLH. Data for test results from HCWs categorised as porters (n = 11), cleaners (n = 91) and 'doctor: other' (n = 98) were excluded from the analysis due to a low sample size in either one or all stages of the pandemic. In total, we analysed 28,909 COVID-19 test results submitted by 4,148 HCWs, of which 772 (3%) tested positive (Table 1).

### Stage of the pandemic, ethnicity and age

The odds of HCWs testing positive for COVID-19 significantly reduced as the pandemic evolved (First wave vs Summer lull: odds ratio (OR) = 2.86; 95% confidence intervals (CI) = 2.25–3.65; p < 0.001; Summer lull Vs Second wave: OR = 3.27; CI = 2.53–4.23; p < 0.001). The HCWs ethnicity was associated with their risk of testing positive, whereby those in the BAME group had higher odds of a positive test result than those of white ethnicity (OR = 1.75; CI = 1.40–2.20; p < 0.001). The odds of testing positive were not significantly associated with age (OR = 0.99; CI = 0.98–1.00; p = 0.113).

### Healthcare worker role

The risk of testing positive for COVID-19 varied between HCW roles (Fig 2A and S3-S5 Tables in S1 Appendix) however, statistically significant differences were only observed for a few contrasts during the summer lull and second wave. Compared to healthcare assistants, the odds of testing positive were lower for allied health professionals (Summer lull: OR = 0.20; CI = 0.05–0.84; p = 0.012), consultants (Summer lull: OR = 0.20; CI = 0.04–0.90; p = 0.023; Second wave: OR = 0.13; CI = 0.02–0.91; p = 0.030) and trainee doctors (Summer lull: OR = 0.12; CI = 0.02–0.69; p = 0.004). The majority of HCW roles had higher odds of a positive test result in earlier stages of the pandemic (S6 Table in S1 Appendix), but a noteworthy exception was healthcare assistants, which was the only role with no significant reduction in the odds of testing positive between the first wave and summer lull (OR = 1.51; CI = 0.78–2.92; p = 0.398).

**Shifts.** During the first wave and summer lull, the risk of a positive test result increased with every additional shift worked, but no significant effect was observed during the second wave (Fig 2B). During the summer lull, a relative increase in the number of night shifts HCWs worked (increasing ratio of night shifts to day shifts worked) resulted in higher risks of testing positive for COVID-19, but no significant effect was identified in the first or second waves (Fig 2C).

### Number of patients

Throughout the pandemic the risk of HCWs testing positive for COVID-19 increased with the number of patients they had contact with (Fig 3A). During the first wave, a relative increase in the number of COVID-19 positive patients contacted by HCWs (increasing ratio of the number of COVID-19 patients seen to the number of patients seen that were not known to have COVID-19) resulted in lower risks of a positive test result (Fig 3B). In contrast, there was no significant effect in the summer lull while, during the second wave, the risk of HCWs testing positive was positively associated with a relative increase in the number of COVID-19 patients they had contact with.

**Table 1. Summary for the healthcare worker population studied during the first year of the COVID-19 pandemic.** A count for the number of healthcare workers and COVID-19 test results are reported. The number and percentage of positive test results are reported. A count and percentage representation (of the observed population) by ethnicity and healthcare worker (HCW) role is also provided.

| | First wave<br>*March—June 2020* | Summer lull<br>*July—Aug 2020* | Second wave<br>*Nov 2020—March 2021* | Overall |
|---|---|---|---|---|
| *Tests* | | | | |
| HCWs | 1,890 | 1,850 | 3,118 | 4,148 |
| COVID-19 tests | 3,454 | 5,570 | 19,885 | 28,909 |
| Positive results | 383 (11%) | 188 (3%) | 201 (1%) | 772 (3%) |
| *Ethnicity** | | | | |
| BAME | 870 (46%) | 905 (49%) | 1,501 (48%) | 2,024 (49%) |
| *HCW role* | | | | |
| Admin | 62 (3%) | 59 (3%) | 94 (3%) | 134 (3%) |
| Allied health professional | 159 (8%) | 108 (6%) | 190 (6%) | 263 (6%) |
| Doctor: consultant | 166 (9%) | 151 (8%) | 282 (9%) | 364 (9%) |
| Doctor: trainee | 155 (8%) | 186 (10%) | 457 (15%) | 582 (14%) |
| Healthcare assistant | 187 (10%) | 231 (12%) | 367 (12%) | 507 (12%) |
| Nurse | 994 (53%) | 973 (53%) | 1,502 (48%) | 1,989 (48%) |
| Other: clinical | 127 (7%) | 102 (6%) | 153 (5%) | 216 (5%) |
| Physiotherapist | 40 (2%) | 40 (2%) | 73 (2%) | 93 (2%) |

* The ethnicity of 66 HCWs was unknown.

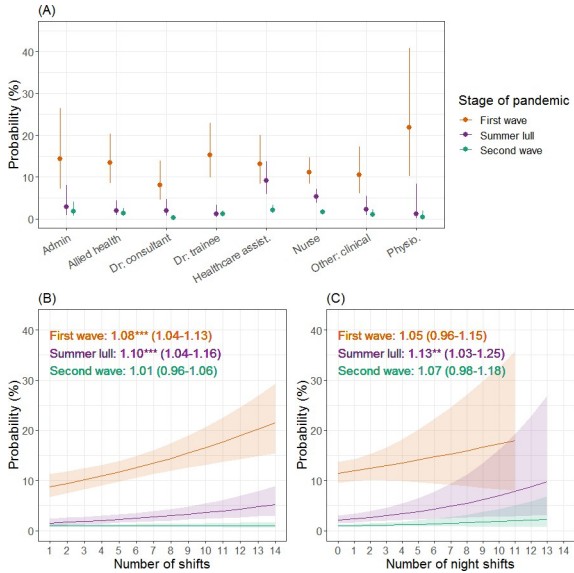

**Fig 2. The probability of a positive COVID-19 test result for different healthcare worker roles and working patterns during the first year of the COVID-19 pandemic.** Estimates (and 95% confidence intervals) are plotted from a mixed effects logistic regression for the first wave (orange), summer lull (purple) and second wave (green). Panel A shows the probability of testing positive for different healthcare worker roles, panel B for the total number of shifts worked two weeks prior to taking a COVID-19 test, and panel C for the relative number of night shifts worked. For plots B & C, the estimated odds ratio, 95% confidence interval and statistical significance (as indicated by asterisks; *p < 0.05; **p < 0.01; ***p < 0.001) are reported.

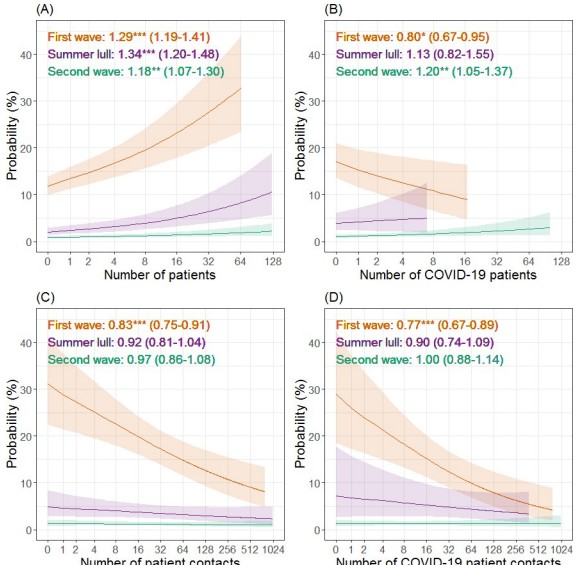

**Fig 3. The relationship between patient contact and the probability of healthcare workers testing positive for COVID-19 during the first year of the COVID-19 pandemic.** Estimates (and 95% confidence intervals) from a mixed effects logistic regression are plotted for the first wave (orange), summer lull (purple) and second wave (green). Plots A & C show the effect of the total number of patients and total number of patient contact events on the probability of a positive test result. Plots B & D show the same effects, but metrics are derived from contact events with only patients identified as COVID-19 positive and is relative to the total number of patients/contacts. Odds ratios, statistical significance (as indicated by asterisks; *p < 0.05; **p < 0.01; ***p < 0.001) and 95% confidence intervals are also reported. For all plots the x-axis is on a logged scale (base 2).

## Number of patient interactions

The total number of patient contact events was negatively associated with the risk of HCWs testing positive for COVID-19 during the first wave, but no significant effect was identified in the summer lull or second wave (Fig 3C). A relative increase in the number of contact events HCWs had with COVID-19 positive patients (increasing ratio of contacts with COVID-19 patients to contacts with patients not know to have COVID-19) was associated with a reduced risk of testing positive during the first wave (Fig 3D). No significant effect was identified during the summer lull or second wave.

## Evidence of activity on floors

When considering whether or not HCWs had evidence of activity on specific floors (at least one patient contact or door event) during the first wave, activity on the majority of COVID-19 floors was associated with increased odds of testing positive compared to when HCWs had no evidence of activity on the focal floor (Fig 4A). The exceptions were floor 8 (respiratory ward) that was not associated with any change in the odds of a positive test result, and floor 3 (critical care) where evidence of activity provided a protective effect. Of the non COVID-19 floors, activity on the ground floor (ED) was associated with higher odds of a positive test result.

During the summer lull and of the COVID-19 floors, evidence of activity was only associated with higher odds of a positive test result on floor 1 (AMU) and floor 10 (CoE). Activity on floor 3 continued to provide a protective effect, as did evidence of activity on some non COVID-19 floors (11, 13, 14 and 16). Higher odds of testing positive persisted for HCWs with evidence of activity on the ground floor. During the second wave, the odds of a positive

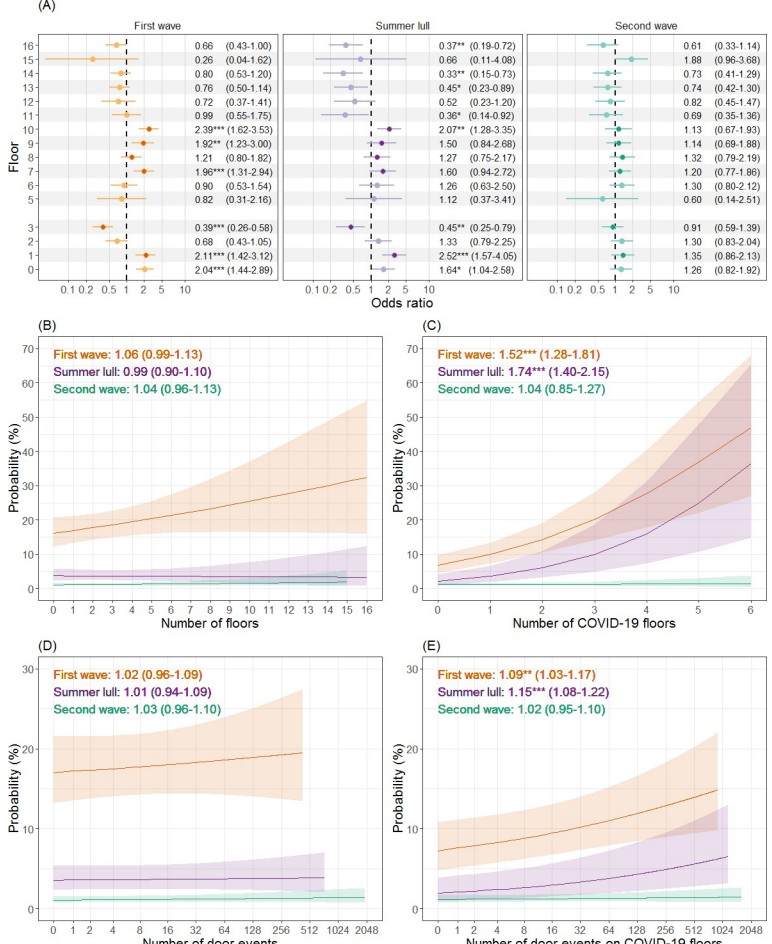

**Fig 4. The relationship between healthcare worker mobility and the risk of testing positive for COVID-19 during the first year of the COVID-19 pandemic.** Odds ratios, statistical significance (*p < 0.05; **p < 0.01; ***p < 0.001) and 95% confidence intervals are estimated from mixed effects logistic regression models, and plotted separately for the first wave (orange), summer lull (purple) and second wave (green). Panel A is a forest plot showing the odds of a positive test when healthcare workers had evidence of activity on a specific floor vs no activity on the floor. Floors that handled the majority of COVID-19 patients (>15%) are identified by darker circles, and floor 4 was excluded due to a lack of clinical spaces. Panel B & C show the estimated probability of testing positive for the total number of floors and COVID-19 floors that individuals were active on respectively. Panels D & E show the estimated probability of testing positive for the number of door events on all floors and on COVID-19 floors respectively. The x-axis of panels D & E are on a logged scale (base 2).

test result were no longer significantly associated with HCWs having evidence of activity on any floor.

## Number of floors

The total number of floors HCWs were active on did not significantly influence the risk of a positive test result in any stage of the pandemic (Fig 4B). However, the spatial context of activity was important, whereby a relative increase in the number of COVID-19 floors HCWs were active on (increasing ratio of the number of COVID-19 floors to the number of non COVID-19 floors) resulted in higher risks of a positive test result during the first wave and summer lull (Fig 4C). No statistically significant effect was found in the second wave.

### Number of door events

The total number of door events logged by HCWs two weeks prior to a COVID-19 test had no significant effect on the odds of HCWs testing positive in any stage of the pandemic (Fig 4D). A relative increase in the number of door events on COVID-19 floors (increasing ratio of the number of door events on COVID-19 floors to the number of events on non COVID-19 floors) resulted in greater risks of HCWs testing positive during the first wave and summer lull (Fig 4E). No significant effect was observed during the second wave.

## Discussion

We have documented the spatial-temporal variation in the risk of SARS-CoV-2 infection for HCWs at a London hospital during the first year of the pandemic. Using routinely collected data we generated simple markers for the within hospital mobility and patient contacts of staff two weeks prior to a COVID-19 test. The association between the infection status of HCWs and their level of patient contact or movement around the hospital was context dependent, and demonstrated significant variations in space and time. Our results show the highest risk of infection among staff was during the first wave of COVID-19 hospital admissions; for HCWs that worked a greater number of shifts, had contact with a greater number of patients and that had higher levels of mobility on and between floors that handled the majority of COVID-19 patients. We also corroborate the findings of previous studies whereby the ethnicity and occupational role of HCWs were identified as important determinants of infection status.

The only temporally consistent behavioural predictor for a HCWs COVID-19 infection status was the total number of patients they had contact with, which had a positive relationship with the likelihood of testing positive. In contrast and during the first wave, HCWs whose work focused more on COVID-19 patients (both in terms of the number of patients and number of patient contacts) were less likely to test positive, but this protective effect was not observed in later stages of the pandemic, with the risk of infection during the second wave increasing the more HCW contacts were focused on COVID-19 patients. The inconsistency of this effect is also reflected in the literature [14,17,24], and our results point towards the importance of considering changing circumstances as the pandemic evolves in time. These circumstances may include factors unobserved in this study, such as shifts in the perception of risk [36], changes to IPC policy and/or challenges relating to IPC activities, such as the supply of PPE or staff shortages [11,13].

HCW mobility within the hospital was a strong indicator for risk of infection and, in line with previous studies, the context of HCW movements was important [2,19,20]. The risk of a positive COVID-19 test result increased with the number of COVID-19 floors HCWs were active on and the number of door events they had on these floors. The number of COVID-19 floors HCWs were active on provides a measure of their exposure to viral hotspots in space, which is intuitively linked to the risk of infection. The relevance of the number of door events is less obvious, but this metric is an indicator for the frequency of movements in and out of COVID-19 hotspots, which may provide a proxy for other high-risk activities such as the need to don and doff PPE or contact with high touch objects [37].

Evidence of activity (at least one patient contact or door event) on a COVID-19 floor was enough to identify HCWs with increased risks of testing positive during the first wave. Contrary to this, activity on two of the COVID-19 floors had no association with increased infection risks; activity on the respiratory ward had no effect and, as reported in other studies, activity on the critical care ward provided protection against infection [22,38]. Given the needs of the patient population on these floors, the layout and facilities would have been better equipped for IPC activities relating to COVID-19 e.g. side rooms to isolate infectious patients,

systems for controlled airflow, appropriate PPE and suitable supplies. The only non COVID-19 floor to be associated with higher infection risks was the emergency department, likely owing to the need to triage patients not yet identified as COVID-19 positive or that were asymptomatic. The spatial variation in the risk of infection became less salient as the pandemic progressed, and was non-existent by the second wave, presumably due to improved IPC policies and activities across the hospital.

Previous studies have found that doctors, nurses and healthcare assistants are more likely to contract COVID-19 than other occupational roles [2,14,17–22,32], and in this investigation healthcare assistants were more at risk than other groups, but this was only after the first wave and in contrast to a few roles. A more conspicuous result was that the number of shifts worked two weeks prior to a COVID-19 test (a proxy for the general exposure of individuals to the hospital environment) had a positive relationship with the likelihood of testing positive. What's more, individuals that primarily worked night shifts during the summer lull were at higher risk of infection. While night shifts have been previously identified as a risk factor [23,31], the reason behind this is unclear. Possible explanations include higher workloads due to lower staffing levels or fewer senior staff to support/supervise IPC activities. The exposure of HCWs to infectious agents will depend on the characteristics of a shift, and shifts will vary in their obligate tasks, therefore investigations into shift profiles for HCW behaviour and the risk of infection would provide further insight into how to better protect staff.

This investigation is not without limitations, one of which is the use of retrospective observational data that may contain sources of bias. Despite using a causal modelling framework and explicitly stating our hypotheses and assumptions (Fig 1 and S1 Appendix), the use of observational data introduces unobserved confounding effects that limit causal claims. We utilised data from the staff testing programme at the hospital, and the testing policy was not consistent throughout the observation period. It is possible that the sample of HCWs taking tests is biased in time and towards those with symptoms as, even when asymptomatic testing policies were introduced in May 2020, tests were taken at the HCWs own discretion. In which case, this bias would be most prominent during the second wave, when vaccinations were introduced. We were also unable to measure an individuals exposure to the community or confirm if infection was acquired through nosocomial transmission (which would require sequencing data). That said, our results are inline with that of previous studies, and we expect infections resulting from community transmission to add noise to the data. Future studies should consider expanding our DAG and, where possible, include data on variables that were not observed in this investigation e.g. PPE use and supply.

A second limitation is in the need to validate behavioural metrics derived from the routinely collected data, as biases may exist due to variations in how HCWs log door events and patient contacts. Studies into the processes underlying the generation of the routinely collected data will help to identify occupational roles that are misrepresented and nuances important for the interpretation of results. In addition, since no centralised electronic rostering database existed, we used the routinely collected data to infer the number of shifts worked, an imperfect method (see supplementary methods in S1 Appendix). However, the rostering database also showed evidence of errors, whereby some shifts labelled as not working had evidence of staff activity in the hospital. It would be preferable to integrate rostering records with data on staff behaviour within the hospital to provide a more accurate measure of worked shifts.

In conclusion, indicators for the within hospital mobility and patient contacts of HCWs can provide insights into the spatial-temporal variations in the risks of infection for staff. These risks will be most pertinent when healthcare systems are perturbed i.e. during outbreaks of disease and the early stages of a pandemic. The relevance of the data sources and models presented in this investigation extend beyond COVID-19, and can be applied to other

communicable diseases (e.g. influenza and norovirus), adapted to consider specific transmission pathways (e.g. particular procedures) and expanded to include data on variables unobserved in this study (e.g. PPE supply). Providing staff testing programmes are in place, digital hospitals have the capability to rapidly assess the infection risk for all staff working on site, in addition to monitoring how risks change between spatially distinct areas of the hospital and in time. Translating these analyses of risk into tools (e.g., apps, dashboards and early warning systems) for routine IPC surveillance will not only help to better protect front-line HCWs and patients, but also in supporting pandemic preparedness.

## Supporting information

**S1 Appendix. Supplemental methods and results.**
(DOCX)

**S2 Appendix. SAFER investigators.**
(XLSX)

## Acknowledgments

For their support we thank the UCLH medical directors Charles House and Gill Gaskin, Pushpsen Joshi at the Joint Research Office, Nathan Lea from the UCLH information governance department, Leila Hail from the UCLH infection control department, Wai Keong Wong, Chris Liddington, Simon Knight, Richard Clarke, David Ramlakhan, David Thompson and Gareth Adams at the UCLH digital services department, Patricia Miralhes and Emily Martyn and all involved with the SAFER research programme.

## Author Contributions

**Conceptualization:** Catherine F. Houlihan, Eleni Nastouli, Ed Manley.

**Data curation:** Jared K. Wilson-Aggarwal.

**Formal analysis:** Jared K. Wilson-Aggarwal, Nick Gotts.

**Funding acquisition:** Catherine F. Houlihan, Eleni Nastouli, Ed Manley.

**Methodology:** Kellyn Arnold.

**Project administration:** Moira J. Spyer.

**Resources:** Moira J. Spyer.

**Writing – original draft:** Jared K. Wilson-Aggarwal, Ed Manley.

**Writing – review & editing:** Nick Gotts, Kellyn Arnold, Moira J. Spyer,
    Catherine F. Houlihan, Eleni Nastouli.

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
