## [Decision Letter · Decision Letter 0]

27 Feb 2023

PONE-D-22-33565Assessing spatiotemporal variability in SARS-CoV-2 infection risk for hospital workers using routinely-collected dataPLOS ONE

Dear Dr. Manley,

Thank you for submitting your manuscript to PLOS ONE. After careful consideration, we feel that it has merit but does not fully meet PLOS ONE’s publication criteria as it currently stands. Therefore, we invite you to submit a revised version of the manuscript that addresses the points raised during the review process.

ACADEMIC EDITOR: The study describes the association between healthcare workers COVID-19 test results and behavioral markers derived from routinely collected hospital data. Congratulations on a detailed and generally well-written paper. The reviewers have provided useful comments to consider before the paper can be accepted for publication. In addition, please pay attention to the typographical errors that exist in the manuscript e.g. line 70 (environment), line 72 (variations).

We look forward to receiving your revised manuscript.

Kind regards,

Mobolanle Balogun

Academic Editor

PLOS ONE

Journal Requirements:

"This study was supported by the UCLH/UCL NIHR Biomedical Research Centre and funding from the UKRI MRC (grant ref: MC_PC_19082), and UCLH Charity. For their support we thank the UCLH medical directors Charles House and Gill Gaskin, Pushpsen Joshi at the Joint Research Office, Nathan Lea from the UCLH information governance department, Leila Hail from the UCLH infection control department, Wai Keong Wong, Chris Liddington, Simon Knight, Richard Clarke, David Ramlakhan, David Thompson and Gareth Adams at the UCLH digital services department, Patricia Miralhes and Emily Martyn and all involved with the SAFER research programme."

"EN, CH and EM were awarded funding from the UCLH/UCL NIHR Biomedical Research Centre and the UKRI MRC (grant ref: MC_PC_19082). Additional funds were awarded to EN from the UCLH Charity, and to EM from the Alan Turing Institute. "

4. Please expand the acronym “UKRI MRC, UCLH/UCL NIHR, UCLH” (as indicated in your financial disclosure) so that it states the name of your funders in full.

6. We note that you have indicated that data from this study are available upon request. PLOS only allows data to be available upon request if there are legal or ethical restrictions on sharing data publicly. For more information on unacceptable data access restrictions, please see http://journals.plos.org/plosone/s/data-availability#loc-unacceptable-data-access-restrictions.

8. PLOS requires an ORCID iD for the corresponding author in Editorial Manager on papers submitted after December 6th, 2016. Please ensure that you have an ORCID iD and that it is validated in Editorial Manager. To do this, go to ‘Update my Information’ (in the upper left-hand corner of the main menu), and click on the Fetch/Validate link next to the ORCID field. This will take you to the ORCID site and allow you to create a new iD or authenticate a pre-existing iD in Editorial Manager. Please see the following video for instructions on linking an ORCID iD to your Editorial Manager account: https://www.youtube.com/watch?v=_xcclfuvtxQ.  

Reviewers' comments:

Reviewer's Responses to Questions

**Comments to the Author**

1. Is the manuscript technically sound, and do the data support the conclusions?

Reviewer #1: Yes

Reviewer #2: Yes

2. Has the statistical analysis been performed appropriately and rigorously? 

Reviewer #1: Yes

Reviewer #2: Yes

3. Have the authors made all data underlying the findings in their manuscript fully available?

Reviewer #1: Yes

Reviewer #2: Yes

4. Is the manuscript presented in an intelligible fashion and written in standard English?

Reviewer #1: Yes

Reviewer #2: Yes

5. Review Comments to the Author

Reviewer #1: I'd like to thank for asking me to review this interesting paper which aims to estimate the degree to which different factors affect the probability of HCWs testing positive for COVID-19 during each of the three identified stages of the pandemic and to demonstrate the epidemiological relevance of deriving markers of staff behaviour from electronic records.

The authors should consider these published papers to deep discuss their results:

doi: 10.3390/ijerph182413053

doi:10.1001/jamanetworkopen.2021.15699

doi: 10.3390/vaccines10122058

Reviewer #2: General comments

1. Congratulations to authors for this very important study that sought to describe the risk of SARS-CoV-2 among health care workers in terms of spatiotemporal variability.

2. Well conducted study with sound methodology

3. Though limited by the retrospective nature of the data, authors used well the data for the study

Specific comments

1. A few typographical errors noted and authors should proofread thoroughly before resubmission eg line 90. Check the spelling of storey

2. As the availability and use of PPEs happens to be a major factor in infection spread, authors should let readers know how healthcare workers use PPE and follow the safety protocols. Would have expected others to include that in the model as an important confounder. Authors can also reference if any studies of the use of PPE.

3. A comment on the presence or otherwise of vaccines during the second wave would also enhance the study.

4. A documentation of the kind of patient contact would also be key. Are they seeing general patients or patients with SARS-COV-2? Will the analysis of that dichotomy enhance the study?

6. PLOS authors have the option to publish the peer review history of their article (what does this mean?). If published, this will include your full peer review and any attached files.

Reviewer #1: No

Reviewer #2: No

---

## [Author Response · Author response to Decision Letter 0]

29 Mar 2023

We would like to thank the editor and two reviewers for taking the time to read and comment on our manuscript. Reference to line numbers are for the revised manuscript with tracked changes. Our response is in bold.

ACADEMIC EDITOR

The study describes the association between healthcare workers COVID-19 test results and behavioral markers derived from routinely collected hospital data. Congratulations on a detailed and generally well-written paper. The reviewers have provided useful comments to consider before the paper can be accepted for publication. In addition, please pay attention to the typographical errors that exist in the manuscript e.g. line 70 (environment), line 72 (variations).

Thank you for the kind words. We have addressed any typographic errors that we have identified.

Reviewer #1

I'd like to thank for asking me to review this interesting paper which aims to estimate the degree to which different factors affect the probability of HCWs testing positive for COVID-19 during each of the three identified stages of the pandemic and to demonstrate the epidemiological relevance of deriving markers of staff behaviour from electronic records.

The authors should consider these published papers to deep discuss their results:

doi: 10.3390/ijerph182413053 

doi:10.1001/jamanetworkopen.2021.15699

doi: 10.3390/vaccines10122058

Thank you for these additional references, we have incorporated the first two into the manuscript.

Reviewer #2

General comments

1. Congratulations to authors for this very important study that sought to describe the risk of SARS-CoV-2 among health care workers in terms of spatiotemporal variability.

2. Well conducted study with sound methodology

3. Though limited by the retrospective nature of the data, authors used well the data for the study

Thank you to the reviewer for the kind comments and for taking the time to review our manuscript.

Specific comments

1. A few typographical errors noted and authors should proofread thoroughly before resubmission eg line 90. Check the spelling of storey

We have now proof read the manuscript and made changes to any identified typographic errors.

2. As the availability and use of PPEs happens to be a major factor in infection spread, authors should let readers know how healthcare workers use PPE and follow the safety protocols. Would have expected others to include that in the model as an important confounder. Authors can also reference if any studies of the use of PPE.

We have added a sentence in the discussion (L392) to say ‘Future studies should consider expanding our DAG and, where possible, include data on variables that were not observed in this investigation e.g. PPE use and supply.’

3. A comment on the presence or otherwise of vaccines during the second wave would also enhance the study.

We detail in the methods (L102) that a ‘mass-vaccination programme began (December 8th 2020)’. We have added a brief sentence in the discussion (L388) to acknowledge the potential bias created from the vaccinations.

4. A documentation of the kind of patient contact would also be key. Are they seeing general patients or patients with SARS-COV-2? Will the analysis of that dichotomy enhance the study?

We agree that the type of patient contact is important. In our analysis we included models to investigate the effect of patient contacts in the context of COVID-19 patients (see methods and results). More nuanced details of the interaction type (e.g. aerosol generating procedure, lines drains and airways etc.) would also be worth investigation as these are in the routine collected data. However, due to the need for further data processing, such an analysis was beyond the scope of the one presented here.

---

## [Editor Report · Decision Letter 1]

3 Apr 2023

Assessing spatiotemporal variability in SARS-CoV-2 infection risk for hospital workers using routinely-collected data

PONE-D-22-33565R1

Dear Dr. Manley,

We’re pleased to inform you that your manuscript has been judged scientifically suitable for publication and will be formally accepted for publication once it meets all outstanding technical requirements.

Kind regards,

Mobolanle Balogun

Academic Editor

PLOS ONE
---

## [Editor Report · Acceptance letter]

13 Apr 2023

PONE-D-22-33565R1 

Assessing spatiotemporal variability in SARS-CoV-2 infection risk for hospital workers using routinely-collected data 

Dear Dr. Manley:

I'm pleased to inform you that your manuscript has been deemed suitable for publication in PLOS ONE. Congratulations! Your manuscript is now with our production department. 

Kind regards, 

on behalf of

Dr. Mobolanle Balogun 

Academic Editor

PLOS ONE